# The Modified Canine Groove Model of Osteoarthritis [note 1]

**DOI:** 10.3390/biomedicines13040913

**Published:** 2025-04-09

**Authors:** Goran S. van der Weiden, Björn P. Meij, Amy van de Belt, Roel J. H. Custers, Sanne K. Both, Marcel Karperien, Simon C. Mastbergen

**Affiliations:** 1Rheumatology & Clinical Immunology, University Medical Center Utrecht, Utrecht University, 3584 CS Utrecht, The Netherlands; 2Department of Orthopedics, University Medical Center Utrecht, Utrecht University, 3584 CS Utrecht, The Netherlands; 3Department of Clinical Sciences, Faculty of Veterinary Medicine, Utrecht University, 3584 CS Utrecht, The Netherlands; 4Developmental BioEngineering, University of Twente, 7522 NB Enschede, The Netherlands

**Keywords:** osteoarthritis, translational, animal model, knee joint, cartilage defect

## Abstract

**Background/Objectives**: In the original canine groove model of osteoarthritis (OA), superficial scratches to the cartilage lead to slow progressive cartilage damage, with inflammation mimicking key aspects of human disease. The present study assesses a modified canine groove model with full-thickness cartilage grooves, gouged with a 3-mm biopsy punch, in the femoral condyles. This modified model enables the study of cartilage repair techniques, such as scaffold implantation. **Methods:** Cartilage defects were induced in the right knee of five *mongrel* dogs (four females, one male; 17 ± 4 months; 25.9 ± 2.0 kg) using the modified groove model, creating two full-thickness cartilage grooves on the femoral condyles. Data of a previously studied cohort of nine dogs (nine females; 18 ± 6 months; 17.6 ± 0.7 kg) with OA induced according to the original groove model served as the canine OA standard. Both groups were monitored up to 45 weeks post-surgery. Pain/function was assessed by force plate analysis, and cartilage integrity, chondrocyte activity, and synovial inflammation were evaluated on the surgically untouched tibial plateaus by macroscopic, histologic, and biochemical analyses. **Results:** Force plate analysis showed no significant changes in either group. Both models exhibited OA features. Experimental knees had more macroscopic and histologic damage, reduced proteoglycan content, and impaired retention of proteoglycans than controls. The modified groove model had less severe cartilage damage and synovial inflammation (*p* = 0.026, *p* = 0.017), with no other significant differences. **Conclusions**: The modified groove model induces OA at a slow pace, mirroring post-traumatic OA development in humans. It represents a mild OA model, comparable to the original groove model, and may be useful for evaluating cartilage repair strategies, such as scaffold implantation.

## 1. Introduction

Osteoarthritis (OA) is a multifaceted condition driven by various factors, leading to a typical pathology characterized by joint degeneration involving articular cartilage, subchondral bone, synovial tissue, menisci, the capsule, ligaments, and muscles [1]. As of 2020, osteoarthritis affected approximately 595 million people globally, representing 7.6% of the population, making it the most common form of arthritis [2]. Disease progression is gradual, often exacerbated by age, and often initiated by a mechanical factor that triggers cartilage degradation in load-bearing areas, sometimes due to trauma [3]. Research is challenged by the slow progression of OA and reliance on surrogate markers, like biochemical markers, and imaging.

To better understand and treat OA, animal models are necessarily used to dissect the complex interactions among genetic, biochemical, and mechanical factors contributing to the disease [4,5]. The canine knee closely resembles the human knee in macroscopic and microscopic anatomy, with similar compartments, cruciate ligaments, menisci, fat pads, and patellar ligaments. Key anatomical differences include the presence of an intra-articular long digital extensor tendon and sesamoid bones (fabellae and popliteal sesamoid) in dogs. Biomechanical differences exist in load transmission, joint congruency, laxity, range of motion, weight-bearing angle, tibial slope, and tibial thrust. Despite these, histological and biochemical properties of cartilage, subchondral bone, synovium, and menisci are well conserved. Notably, both species experience comparable spontaneous pathologies, such as cruciate ligament deficiency, meniscal injuries, and osteochondrosis. This similarity makes dogs superior to smaller species for translational OA research, where anatomical and biochemical differences are more pronounced [4,5,6,7,8].

The canine ACLT (anterior cruciate ligament transection) model of OA has been fundamental in studying OA’s development and progression, as well as its treatment effects, including the use of NSAIDs (non-steroidal anti-inflammatory drugs), corticosteroids, and disease-modifying OA drugs (DMOADs) [4,5]. These treatments, however, have associated side effects. The long-term use of NSAIDs has been associated with gastrointestinal, renal, and cardiovascular risks [9], while repeated intra-articular corticosteroid injections may accelerate cartilage degeneration and impair joint homeostasis [10]. Additionally, many candidate DMOADs have shown limited efficacy in clinical trials and are often accompanied by systemic toxicity or off-target effects, restricting their therapeutic applicability [11]. The ACLT model features a permanent trigger, joint instability, that may complicate cartilage treatment effects because attempts to repair are counteracted by continuing joint instability [4,5].

The canine groove model has been developed to overcome this limitation of the ACLT model [12]. In this model, OA is induced by surgically applied chondral damage to the weight-bearing cartilage of the femoral condyles, followed by mechanical loading. It effectively mimics primary chronic cartilage trauma, a common precursor to joint degeneration in humans. Unlike other models, it exhibits progressive cartilage degeneration over time, alongside low grade synovial inflammation [13]. There is an initial post-surgical increase in synovial inflammation, which subsides over time but does not return entirely to baseline [13]. Degenerative changes continue up to 40 weeks post-surgery [14], with cartilage integrity and matrix turnover resembling those observed in human clinical OA [15]. In terms of pain and functional impairment, the procedure demonstrates that animals exhibit lameness and reduced joint function following the induction of OA using the groove model. These clinical signs align with the degenerative changes observed histologically, reinforcing the model’s validity in mimicking human OA symptoms [16]. However, although the model is progressive until 40 weeks post-operatively, development of full-blown osteoarthritis, as is shown for the ACLT model at 54 months post-surgery [12], remains to be demonstrated.

The dominance of cartilage damage in this model provides a clearer assessment of a treatment’s direct effect on cartilage, minimizing indirect effects from potential anti-inflammatory properties of treatment. Notably, this model avoids the ongoing effects of permanent triggers like joint instability, which could diminish treatment efficacy. As a result, the groove model is considered to be particularly well-suited for evaluating the effectiveness of structure-modifying or disease-modifying OA treatments based on the premise that cartilage repair is feasible.

Acute trauma can lead to well-defined areas of localized cartilage loss known as cartilage defects, which are considered one of the major contributors of OA in the knee joint. These defects differ from the more diffuse chondral damage induced by the original groove model [12,13,14,16], where cartilage is scratched, with full-thickness damage occurring only along the groove’s tip. In contrast, cartilage defects affect the full thickness of a larger cartilage area.

Currently, there is a lack of suitable models for adequately evaluating cartilage repair strategies aimed at filling or regenerating these defects by techniques such as using cell-free polymer-based hydrogels as scaffolds for cartilage regeneration [4,5,8]. Our goal was to develop a model that allows for the assessment of such repair strategies while maintaining key characteristics of OA observed in the original groove model. As such, we have developed the adjusted groove model to more closely resemble these defects and allow for the evaluation of suitable repair techniques.

## 2. Materials and Methods

### 2.1. Animals

#### 2.1.1. Original Groove Model

Data from a previously published cohort of 9 *mongrel* dogs [17] (9 female; mean ± SD age of 18 ± 6 months (radiographic confirmation of growth plate closure, absence of pre-existing conditions that affect the outcomes or analysis, such as OA or fractures); and a mean ± SD weight of 17.6 ± 0.7 kg) was used as a retrospective comparison. The Utrecht University Committee for Experiments on Animals approved the study according to Dutch law (DEC no. 2007.III.02.029).

#### 2.1.2. Modified Groove Model

Five skeletally mature *mongrel* dogs (4 female, 1 male) were obtained from the Central Laboratory Animal Research Facility of the Utrecht University (mean ± SD age of 17 ± 4 months (radiographic confirmation of growth plate closure, absence of pre-existing conditions that affect the outcomes or analysis such as OA or fractures) and a mean ± SD weight of 25.9 ± 2.0 kg). Originally, the modified groove model consisted of n = 6; however, one female dog was terminated prematurely due to unrelated illness and was therefore excluded. The Utrecht University Committee for Experiments on Animals approved the study according to Dutch law (DEC no. AVD11500202010905).

#### 2.1.3. Housing

Animals of both cohorts were housed in small groups (2–3 dogs per 3 × 4 m area) and were exercised in groups on a larger open patio (6 × 8 m) for at least 2 h each day during the entire experiment. They were fed a standard diet and given water ad libitum.

### 2.2. Induction of Joint Degeneration

#### 2.2.1. Original Groove Procedure

In the original groove model, OA was induced in the right knee joint by an experienced surgeon, as described previously [12,13,14,17]. Through a mini-open medial parapatellar incision, with the knee flexed to expose the weight-bearing region of both femoral condyles, 10 grooves were made only on the weight-bearing regions of the femoral condyles using a Kirschner wire (Stryker; Kalamazoo, MI, USA); 1.5 mm diameter, bent to 90° at 0.4 mm from the tip; Figure 1). This ensured that the depth of the grooves was restricted to the cartilage depth to prevent damage to the underlying subchondral bone. The resulting cartilage fragments were small (<2 mm) and sometimes still connected to the surrounding cartilage and, therefore, left in the knee joint.

#### 2.2.2. Modified Groove Procedure

In the modified groove model, OA was induced in the right knee joints of all 5 dogs by an experienced surgeon. Using an open medial parapatellar approach, the knee joint was accessed through an incision. The patella was gently mobilized laterally, and the knee was maximally flexed to expose the weight-bearing region of both femoral condyles. Two full-thickness cartilage grooves were made in the weight-bearing region of each femoral condyle using a 3 mm biopsy punch as a gouge, whilst ensuring the underlying subchondral bone remained undamaged (Figure 1). The grooves were created parallel to the directional movement of the femoral condyles to ensure their placement on the weight-bearing area of the femoral condyles. The obtained cartilage pieces were large (up to 2 cm) and were not connected to the cartilage. Due to foreseen interference, we removed the fragments from the joint.

#### 2.2.3. Perioperative Management

For both models, perioperative management of the dogs was identical.

Prior to all procedures, dogs were administered general anesthesia, starting with premedication using dexmedetomidine and butorphanol intramuscularly, followed by induction with propofol intravenously. They also received pre-emptive analgesia with buprenorphine and carprofen, and antibiotic treatment with cefazoline, all administered intravenously. All procedures were conducted under sterile conditions with continuous monitoring of vital signs, including heart rate, respiration, and oxygen saturation. Anesthesia was maintained through the inhalation of isoflurane, adjusted between 1.0% and 1.5%. The anesthetic protocol was experiment- and dog-specific, and created in consultation with the local veterinary team, similar to previously published experiments [16].

Importantly, for both models, when applying grooves to the femoral condyles, the menisci and tibial plateaus were left untouched. The joint capsule and skin were sutured according to their anatomic layers. The contralateral (left) knee joint was left completely untouched and served as an internal control. Previous studies confirm its suitability, showing no significant differences between healthy control joints [18]. Using an internal control reduces inter-animal variation, minimizing the number of animals required per experiment.

### 2.3. Outcome Measures

#### 2.3.1. General

The assessment of both models utilized previously described outcome measures [17], including force plate analysis, cartilage tissue integrity, chondrocyte activity, and synovial tissue inflammation. Cartilage tissue integrity and chondrocyte activity were assessed on the untouched tibial plateaus so as to minimize influence of the direct applied surgical damage on the femoral condyles [17]. A 45-week follow-up period was chosen to allow adequate time for OA progression and to facilitate comparisons with the retrospective cohort. This duration also corresponds to the standard timeframe for implementing regenerative therapies, aligning with the intended purpose of this canine model.

#### 2.3.2. Force Plate Analysis

As a surrogate measure of pain/functional ability, limb loading during gait was evaluated by force plate analysis, and values were measured using a floor-mounted force plate (0.6 × 0.9 m) set into a path 13 m long. A type Z4852C force plate, type 1681 cables, 5007 charge amplifiers, and a type 5217 summation amplifier (Kistler; Winterhur, Switzerland) were used. The sampling rate was 100 Hz [19]. The dogs were guided to walk at a constant walking speed: both the front and hind leg of the same laterality must be placed subsequently on the force plate mounted flush to the surface. The force plate measured peak ground reaction forces in the 3 dimensions (x, y, and z). A single handler guided the dogs by leash over the force plate at a constant walking speed of ~1 ± 0.2 m/second (mean ± SD). A successful run consisted of sequential, distinct paw strikes of the left and right hind limbs. On average, 10 valid runs of each side were collected, and ground reaction forces were averaged for each limb separately. The measured ground reaction forces were normalized to the body weight and expressed in newtons per kilogram. Maximum stance force (Fz) and braking force (Fymax) values were used for analyses, as these correlate best to the clinical features of OA in dogs [20]. Measurements were obtained after full surgical recovery, at weeks 35 and 45.

#### 2.3.3. Cartilage Integrity

Macroscopic cartilage damage of the surgically untouched tibial plateaus was evaluated using the Osteoarthritis Research Society International (OARSI) canine scoring system [6]. High-resolution images of the cartilage were anonymized and randomized (GW) and independently scored by two observers blinded to the origin of the image (AB, SCM). The score is assigned based on the following criteria and is determined by the most severe pathology noted: 0 = smooth surface, 1 = roughened, 2 = slightly fibrillated, 3 = fibrillated, 4 = damaged. Scores of the two observers were averaged (maximum of 4). This score was used as the representative score for each photograph and was used for statistical analysis.

Cartilage samples for histological and biochemical analyses were obtained from predetermined locations on the weight-bearing areas of the tibial plateau of both the operated and control joints. This was done similarly to all animals [12].

Histological cartilage sections (5 µm thick) were stained with safranin-O/fast green with Weigert’s hematoxylin as a counterstain for detailed examination [21,22]. These samples were also anonymized and randomized (GW) and scored by the same observers who were blinded to the origin of the samples (AB, SCM) according to the OARSI canine scoring guidelines (maximum score of 36) [6].

Biochemical analysis involved the quantification of glycosaminoglycan (GAG) content, which is indicative of cartilage tissue proteoglycan levels, from eight predefined samples across the tibial plateaus [23,24]. GAGs were extracted using papain (P-3125; Sigma–Aldrich; Saint Louis, MO, USA; 25 mg/mL in 50 mM phosphate buffer, pH 6.5, containing 2 mM N-acetyl cysteine and 2 mM Na2-EDTA), precipitated, and then stained with Alcian blue (Alcian blue 8GX, A-5268; Sigma–Aldrich; Saint Louis, MO, USA; saturated in 0.1 M sodium acetate buffer, pH 6.2, containing 0.3 M MgCl_2_; 30 min, 37 °C), and measured spectrophotometrically at an absorbance of 620 nm, with chondroitin sulfate as a reference (C4383; Sigma–Aldrich; Saint Louis, MO, USA). The results were normalized to the wet weight of the cartilage explants, expressed in mg/g [13].

#### 2.3.4. Chondrocyte Activity

To assess the retention of newly formed proteoglycans (PGs), a 3-day release study of pulse-labeled PGs was conducted using ^35^SO_4_^2^ (Na_2_^35^SO_4_, 14.8 kBq/200 μL, carrier-free; NEX-041-H; DuPont Wilmington, DE, USA) as a tracer. Labeled GAGs were precipitated from both a papain digest of the tissue and the culture medium using Alcian blue [24]. The rate of sulfate incorporation was normalized based on the specific activity of the medium, the 4-h labeling duration, and the wet weight of the explants. The release of these newly formed PGs was quantified as a percentage of the newly synthesized proteoglycans.

Furthermore, to calculate the overall loss of GAGs (including both resident and newly formed) over the three days, the Alcian blue staining of the medium was measured spectrophotometrically at an absorbance of 620 nm. The total GAG release was then expressed as a percentage of the original content in the cartilage tissue, providing a comprehensive view of PG retention and loss [13].

#### 2.3.5. Synovial Tissue Inflammation

Macroscopic inflammation was graded using high-resolution digital photographs of the synovial tissue, analyzed according to the OARSI canine scoring system for synovial inflammation, with a maximum score of 5 [6]. This grading was performed in a random order by two observers who were blinded to the origin of the images (AB, SCM). The severity of inflammation was graded for overall color, angiogenesis, and fibrillation: 0 = none, 1 = slightly, 2 = strong, 3 = moderate, 4 = marked, 5 = severe. The individual scores were averaged for the two observers (a maximum of 5) and were used as the representative score for each joint for statistical analysis.

Histological analysis of synovial tissue inflammation was conducted on HE-stained sections, following the canine OARSI scoring system for synovial inflammation, with a maximum score of 18 [6]. This assessment was carried out in a random order by two blinded observers (AB, SCM).

#### 2.3.6. Calculations

Force plate analysis utilized the average of 10 runs to determine the mean peak ground reaction forces for both hind limbs. The mean values at weeks 35 and 45 after surgery were averaged for a single outcome for each limb separately.

For macroscopic evaluation, the scores for cartilage and synovial tissue provided independently by the two blinded observers were averaged to provide a single representative value for each sample.

For histological assessments, four cartilage samples from the tibial plateau and three synovial tissue samples per joint, each independently provided by the two observers, were averaged.

For the biochemical analysis of GAGs by the Alcian blue assays, as described above, the average of eight cartilage samples per tibial plateau were used for the proteoglycan data of each joint.

### 2.4. Statistical Analysis

The contralateral joint was used as an internal control. Nonparametric comparisons between the experimental and contralateral control joints were made by using Wilcoxon’s signed rank test. The absolute differences between experimental and contralateral control joints (delta changes) were compared between the two models (n = 9 and n = 5) using a Mann–Whitney U test. p-values less than 0.05 were considered statistically significant.

## 3. Results

Shortly after surgery, all animals were fully active, with subjectively normal joint loading and movement. During the whole experiment, no adverse events were reported.

### 3.1. Force Plate Analysis

To gain clinically related data on pain and function (loading), force plate analysis was performed. In both the original and the modified groove model, force plate analysis showed no statistically significant differences in the stance and braking force at week 35–45 post-surgery between the experimental and contralateral limbs (Figure 2, Table 1). The differences between both models were not statistically significant.

### 3.2. Cartilage Integrity

Comparisons of the surgically untouched tibial plateaus of the right experimental knee joints and the contralateral left internal control knee joints revealed statistically significantly more macroscopic damage for the experimental limbs in both the original groove model and the modified groove model (*p* = 0.007 and *p* = 0.034, respectively; Table 1). The modified groove model showed less cartilage damage in the surgically untouched tibial plateau than the original groove model (*p* = 0.026; Figure 3).

Comparable results were found for the histologic assessments of the original and modified groove models. Both showed statistically significantly enhanced scores, with *p*-values of 0.008 and 0.043, respectively (Figure 3). The difference in histologically assessed damage between both models was not statistically significant.

Generally, histological observations were biochemically confirmed. In the original groove cohort, the proteoglycan content was significantly lower in the experimental joints compared to the control joints (*p* = 0.008; Table 1, Figure 4). In the modified groove model, the loss of proteoglycan content was not statistically significant (*p* = 0.225; Table 1, Figure 4). There was no statistically significant difference in the PG content change between both models.

The increased release of newly formed PGs was detected in both the original and modified groove models (*p* = 0.015 and *p* = 0.043, respectively), indicating diminished retention of these newly formed proteoglycans in the cartilage matrix (Table 1, Figure 4). Both models showed a similar change in the release of newly formed PGs.

The proteoglycan content, total PG released, and release of newly formed PGs are indicative of degeneration in the tibial cartilage, and support the macroscopic and histological findings.

The total loss of proteoglycans was statistically significantly increased in both the original (+6.2%, *p* = 0.021) and modified groove models (+3.3%, *p* = 0.043; Table 1, Figure 4). Again, no statistically significant differences between models were found.

### 3.3. Synovial Inflammation

The macroscopic synovial inflammation scores were significantly higher for the experimental limbs compared to the contralateral controls in both the original and modified groove models, with *p*-values of 0.007 and 0.039, respectively (Table 1, Figure 5). The modified groove model exhibited less severe macroscopic synovial tissue inflammation compared to the original groove model (*p* = 0.017; Figure 5).

Histologically, both the original and modified groove cohorts showed a statistically significant, but limited, increase in synovial tissue inflammation, with *p*-values of 0.008 and 0.043, respectively (Table 1, Figure 5). Histologically, the difference between the two models was not statistically significant (Figure 5).

## 4. Discussion

Features of OA in the canine groove model resemble those of mild to moderate OA in humans. The original groove model has been used to assess treatments aimed at slowing down the damage progression and even repairing the joint [4,5,8]. Specifically for knee joint distraction, improvements in the tissue structure obtained with this model corroborated the observed cartilage repair activity in human clinical studies observed by MRI (magnetic resonance imaging), radiography, and analysis of biochemical markers [25]. This strong translational value of the groove model makes it an ideal model to expand upon. Unfortunately, because of the generalized cartilage damage introduced by diffuse scratching, the model is less suitable for evaluating the efficacy of repair strategies based on the filling of focal defects. Therefore, the present study explores a modified version of the groove model, introducing two well-defined full-thickness cartilage defects per femoral condyle, offering a more controlled and reproducible approach to inducing OA-related degeneration resulting from focal cartilage defects. Also, this model leads to OA development in the opposing, surgically untouched tibial plateau, which is in line with the original groove model and with existing literature [12,13,14,17]. This progression occurs gradually over time, mirroring the slow onset of OA in humans, which can take years to fully manifest.

A key distinction between both models lies in their degrees of variability and response to the applied surgical damage. In most analyzed parameters, the modified groove model tends to show a lower degree of degeneration than the original model. Additionally, the modified groove model exhibits less variation, likely due to the greater standardization in creating focal-like defects. The increased consistency and larger continuous surface area of the cartilage defects has an advantage over the original groove model in that it makes the modified groove model more suitable for testing cartilage repair therapies, such as defect filling using biomaterial-based scaffolds or other regenerative treatments.

By bridging the gap between isolated focal defects and more extensive cartilage damage, the modified groove model provides a reproducible and clinically relevant platform for investigating regenerative therapies. Its improved standardization and applicability to cartilage repair strategies makes it a promising tool for advancing pre-OA interventions to prevent post-traumatic osteoarthritis, such as cartilage repair strategies aimed at filling or regenerating these defects by using cell-free polymer-based hydrogels as scaffolds for cartilage regeneration.

Intensified loading (i.e., due to malalignment, removal of meniscus, joint instability, external forces) seems to be a contributing factor to the development of OA after cartilage is damaged [4,26]. Small cartilage defects do not necessarily cause widespread changes in the knee joint, especially when not directly loaded (i.e., the non-weight-bearing area of the knee or protection from overlaying menisci) [4,27,28]. The modified and the original groove model show limited pain/disability as measured by the force plate analysis. As cartilage tissue is not innervated, this could be due to limited general damage to the joint, which is important for this degenerative model, with limited inflammatory activity. Though both models did not differ, the modified groove model seemed less sensitive. This might be explained by the fact that the obtained cartilage from the grooves was removed completely (as opposed to the original groove model) when creating the grooves, as loose bodies are known to cause pain and irritation [29]. In the original groove model, the remaining cartilage material was not removed after grooving, whereas in the modified groove models, the loose bodies were removed because of their size, which we expected could cause a more severe and more variable response to these loose bodies.

The removal of the cartilage of the grooves in the modified groove model potentially eliminated an additional continued contributing factor to the degeneration of the joint, as would be the case in the original groove model, where the cartilage residuals of the grooves were left in the joint. An advantage in the modified groove model compared to other models is the absence of additional continuing triggers for OA, apart from the cartilage damage itself, allowing for a more accurate assessment of treatment options.

Biochemical and histological analyses were deliberately performed on the surgically untouched opposing tibial cartilage. It has been described that damage found on the tibial plateau in the original groove model was most likely also the result of incongruities on the articular surfaces after the condyles were damaged [13]. However, in the modified groove model, this incongruity is not as pronounced; this might be due to the already-mentioned removal of residual cartilage pieces. An additional reason might be that the large grooves were created parallel to the directional movement of the femoral condyles, making smoother movement possible. This was done to ensure the grooves could be made upon the weight-bearing area of the femoral condyles: if placed on the non-weight-bearing area, there would be no development of OA due to lack of loading. This is opposed to the original groove model, where a more disorganized pattern is created with smaller but rougher grooves. This is reflected in the macroscopic cartilage tissue damage score, which seemed to be more affected in the original groove model. Nonetheless, both models showed statistically significant macroscopic and histologic cartilage deterioration in the treated joint compared to the contralateral control joint. This difference was also found in the retention of newly formed proteoglycans and in the loss of PGs, indicating a similar process of OA development in the two models.

The inflammation of synovial tissue was mild in both the modified and original groove cohorts, as described previously for the canine groove model. Macroscopic synovial tissue inflammation was more severe in the original groove model (*p* = 0.017; Figure 3). As synovial tissue inflammation can be triggered by loose cartilage [29], this could be due to still-present loose bodies in the original groove model compared to the modified model. Though in histological analysis, no statistically significant difference was shown between both groups.

### Limitations

This study has several limitations that warrant consideration. First, although force plate analysis is a well-established method for evaluating functional impairment, it is inherently sensitive to variability—particularly in canine models, where enforcing consistent gait patterns is challenging. As a result, subtle changes in OA progression or functional disability may be obscured by natural fluctuations in gait dynamics, limiting the sensitivity to detect longitudinal trends or differences between models [30,31].

Second, this study relied on a retrospective comparison between the modified and original groove models. While this approach was chosen to reduce animal use and associated costs—and the historical data were derived from an identical experimental setup (including the same surgical team, animal supplier, housing conditions, and analytical protocols)—the retrospective nature of the comparison reduces control over potential confounding variables and temporal changes that may have occurred between study periods.

Third, the sample size for the modified groove model was relatively small (n = 5), which may limit statistical power, potentially masking meaningful differences between groups.

Lastly, the biochemical and histological analyses focused exclusively on tibial cartilage, omitting evaluation of other joint tissues, such as the menisci, subchondral bone, and ligaments, which also play critical roles in OA pathology and pain mechanisms [32,33].

These limitations should be considered when interpreting the current findings. Further validation is necessary to fully establish the utility of the modified groove model.

## 5. Conclusions

The modified groove model demonstrates early features of osteoarthritis comparable to the original groove model, leading to similar structural damage and inflammation, although macroscopically less pronounced. Unlike models that focus solely on isolated cartilage defects or advanced OA, this modified groove model represents an intermediate stage of post-traumatic OA, bridging the gap between focal cartilage injuries and widespread joint degeneration. As such, it provides a valuable platform for testing pre-OA interventions, particularly cartilage repair strategies, making it a promising tool for advancing regenerative treatments.

## Figures and Tables

**Figure 1 biomedicines-13-00913-f001:**
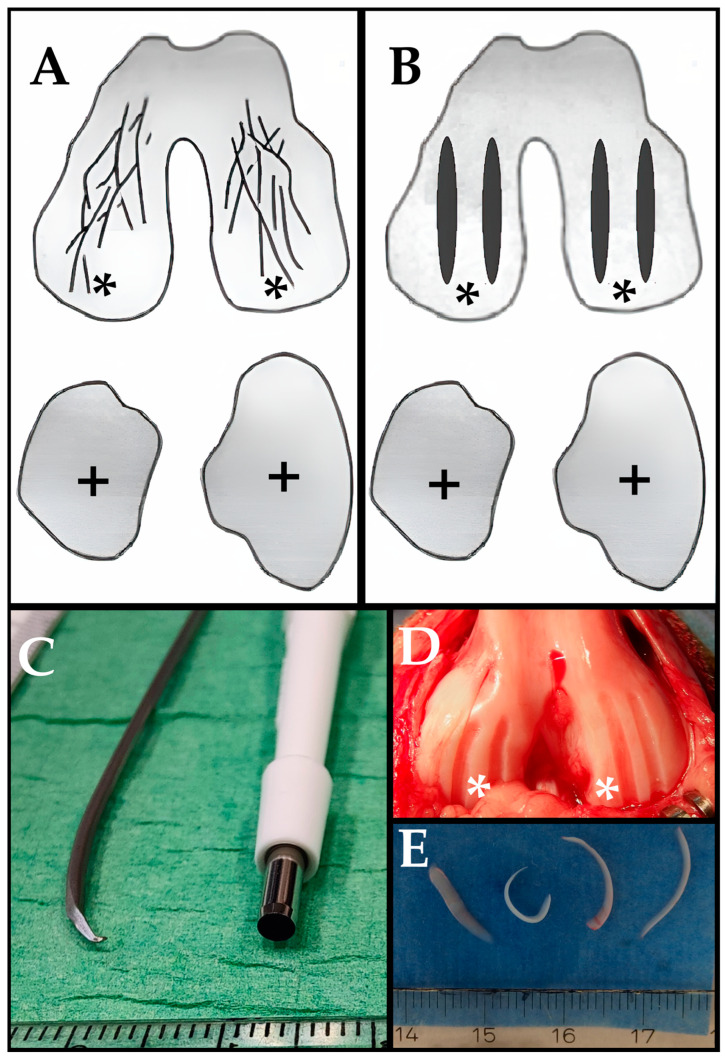
Illustration of the applied surgical damage. (**A**) Original groove model; 10 randomly linear grooves were made on the femoral condyles using a 1.5 mm Ø Kirschner wire bent to 90° at 0.4 mm from the tip. (**B**) Modified groove model; 2 full-thickness grooves of 2–3 mm wide were made on the femoral condyles using a 3.0 mm biopsy punch as a gouge. (**C**) Photograph of the Kirschner wire (left side) and biopsy punch (right side). (**D**) Photograph of the modified grooves directly after the application. (**E**) Removed cartilage pieces. *: Femoral condyle. +: Tibia.

**Figure 2 biomedicines-13-00913-f002:**
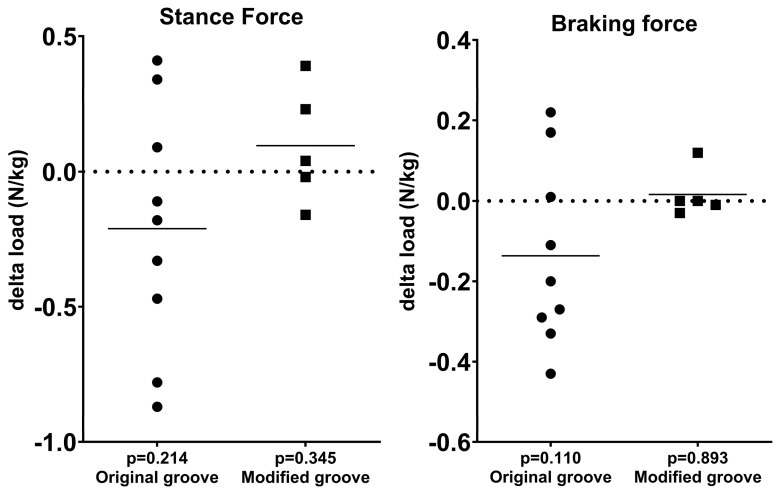
Force plate analysis: differences in stance force and braking force between experimental and contralateral control limbs in each animal at week 35–45 post-surgery are shown; *p*-values for the differences are given at the bottom. Each symbol represents a single animal; horizontal lines show the mean. N/kg = Newtons/kilogram.

**Figure 3 biomedicines-13-00913-f003:**
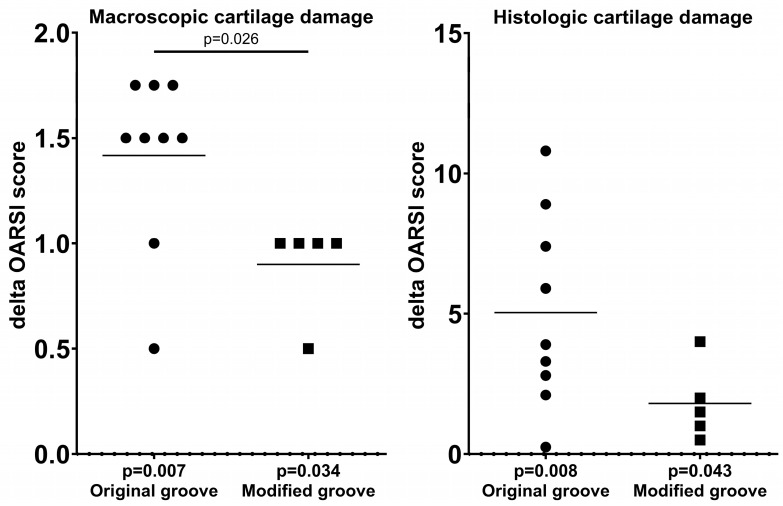
Macroscopic and histologic assessments of the surgically untouched tibial cartilage according to the OARSI canine scoring system. Differences in scores between experimental and contralateral control joints in each animal are shown, as well as the average scores. *p*-values are given per model, as well as for differences between the models in cases of statistical significance. OARSI = Osteoarthritis Research Society International.

**Figure 4 biomedicines-13-00913-f004:**
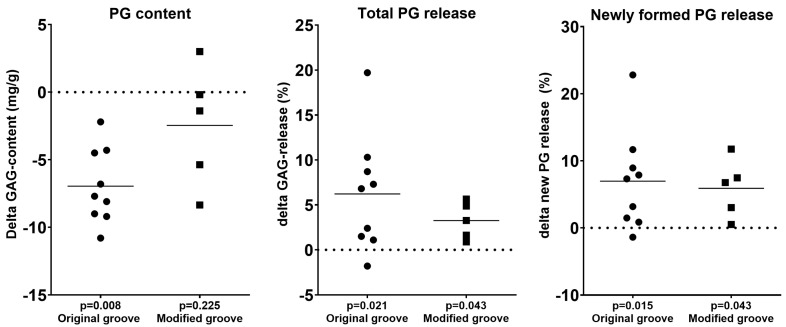
Differences between the experimental and contralateral control joints in glycosaminoglycan (GAG) content as a measure of proteoglycan (PG) content, release of newly formed PGs as a measure of retention of these newly formed PGs, and total PG released over 3 days. The differences in scores between the experimental joint and the contralateral control joint in each animal are shown; *p*-values for the differences are given at the bottom. Each symbol represents a single animal; horizontal lines show the means. The differences between the models were not statistically significant.

**Figure 5 biomedicines-13-00913-f005:**
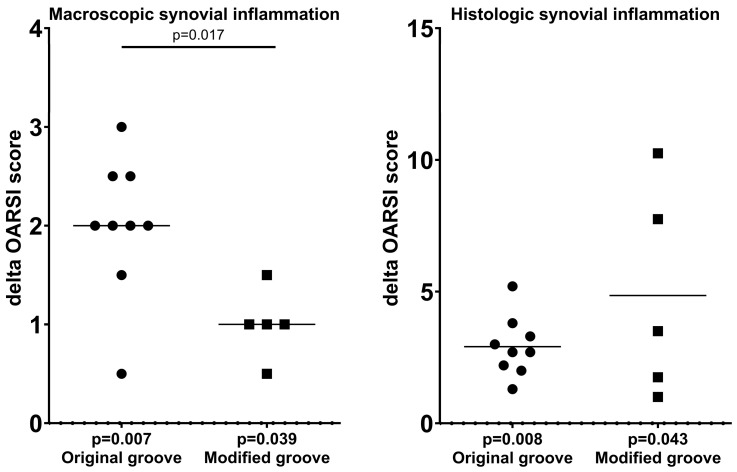
Macroscopic and histologic assessments of the synovial tissue according to the OARSI canine scoring system. Differences in scores between the experimental and contralateral control joints in each animal are shown, as well their average scores. *p*-values are given per model as well as for differences between the models, when statistically significant. OARSI = Osteoarthritis Research Society International.

**Table 1 biomedicines-13-00913-t001:** Absolute values of the experimental and control knee joints for all parameters for the original and the modified groove models.

	Original Groove Model			Modified Groove Model		
Outcomes	Experimental	Control	*p*	Experimental	Control	*p*
Stance force	4.0 ± 0.5	4.2 ± 0.3	0.214	3.9 ± 0.3	3.8 ± 0.2	0.345
Braking force	0.6 ± 0.1	0.8 ± 0.2	0.110	0.6 ± 0.2	0.6 ± 0.1	0.893
Macroscopy of cartilage	1.5 ± 0.5	0.1 ± 0.2	**0.007**	0.9 ± 0.2	0.0 ± 0.0	**0.034**
Histology of cartilage	9.4 ± 2.3	4.4 ± 1.7	**0.008**	6.1 ± 2.2	4.3 ± 1.3	**0.043**
Macroscopy of synovial tissue	2.1 ± 0.7	1.0 ± 0.4	**0.007**	1.0 ± 0.4	0.0 ± 0.0	**0.039**
Histology of synovial tissue	4.1 ± 1.5	1.1 ± 0.6	**0.008**	9.6 ± 3.0	4.7 ± 2.9	**0.043**
PG content	27.8 ± 3.3	34.8 ± 2.9	**0.008**	24.7 ± 6.9	27.1 ± 8.6	0.225
Newly formed PG released	50.2 ± 10.4	43.2 ± 4.9	**0.015**	39.8 ± 14.1	33.9 ± 11.0	**0.043**
Total PG released	27.7 ± 7.6	21.5 ± 2.9	**0.021**	19.0 ± 4.6	15.8 ± 4.1	**0.043**

Mean values ± standard deviations are presented. Significant *p*-values are in bold. PG = proteoglycan.

## Data Availability

The data presented in this study are available on request from the corresponding author, access is restricted due to ethical restrictions.

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
