# Peer review of "The Modified Canine Groove Model of Osteoarthritis [Author-notes fn1-biomedicines-13-00913]"

_biomedicines, 2025, doi:10.3390/biomedicines13040913_

Round 1
Reviewer 1 Report
Comments and Suggestions for Authors
Biomedicines-3509642 (MDPI)
Type
Article
Title
The modified canine groove model of osteoarthritis.
Comments:
Comments in each section are as follows
Abstract: Information about earlier published article is not necessary in the abstract. Name of the methods used to evaluate needful parameters in the study must be mentioned.
Results must be crisp in the abstract section. Detail results must be provided in the result section of main manuscript and not in abstract
Introduction:
- Prevalence of disease must be mentioned
- Side effects of NSAIDs, corticosteroids, and DMOADs must be mentioned along with references
- Fig of Canine groove model showing surgically applied chondral damage to the weight-bearing cartilage of the femoral condyles may be generated and must be labelled appropriately for information and clarity in understanding.
- Mention anatomical differences between canine and human knees.
- How this model will be utilized for testing regenerative therapies. Please explain sufficiently
- What do the matrix turnover means that resembles human clinical OA. Incorporate information along with reference
- Why progression to fully developed OA, in the ACLT model at 54 months post-surgery, takes more time? How much time ACLT model takes and how much time groove model Sufficient information must be provided along with references
Further, it is mentioned (line 63) that groove model exhibits progressive cartilage degeneration alongside a gradual reduction in synovial inflammation. Its highly confusion. How reduction in synovial inflammation happens when there is progressive cartilage degeneration? Clarification needed.
- What is diffuse chondral damage. Incorporate information along with references
- Line 49, References for the line “The dog is one of the best … knee OA” must be incorporated.
- Line 56, References for the line “This ACLT model, …….joint instability” must be incorporated.
- Reference for line 82 “Currently, there is a lack of suitable models for adequately evaluating cartilage repair strategies” must be mentioned.
Methods:
- Which method was used for animal sampling and what was the duration of experiment using the said models. Mention it.
- It is mentioned line 91 that the data of Original groove model (ref 10) was used as retrospective comparison. What were these data that were used to compare the current results.This data got published in the year 2015. Please clarify.
- It is mentioned, line 115 that the resulting cartilage fragments were left in the knee joint.Further, in modified groove model, the obtained cartilage pieces were removed from the joint. Why so? Is there any specific reason. Please clarify.
- How Maximum stance force (Fz) and braking force (Fymax) values were measured. Name the instrument, company name etc.
- What is scaffold implantation? Incorporate sufficient information along with references
- Briefly mention the Safranin- O/Fast green staining procedure, Alcian blue staining procedure, HE-stained, sections quantification of glycosaminoglycan (GAG) content, measurement of cartilage tissue proteoglycan levels etc along with references.
- Mention company and country name for purchasing chemicals (such as papain, Alcian Blue, chondroitin sulfate etc), instruments (such as spectrophotometer, microscope etc)
- References for dogs anesthesia must be incorporated.
- Mention why groups were monitored up to 45 weeks post-surgery. Incorporate information along with references
- Mention why the results were analysed 35 and 45 days after post surgery
- What is nonparametric comparisons
- How biochemically proteoglycan content was analysed. Briefly describe
- Mention the importance of Mann-Whitney U test and Wilcoxon's signed rank test
Outcome measures :
- Briefly mention how and why Force plate analysis were performed along with references
- Mention the criteria used macroscopic evaluation, the scores for cartilage and synovial tissue
- Name the biochemical assays performed using eight cartilage samples per tibial plateau and how proteoglycan data of each joint was obtained.
- What is contralateral joint and why it was used as internal control
Results:
- What is stance and braking force and why these parameters were measured.Importance of these parameter in this study must be mentioned.
- In Cartilage integrity assay, mention surgery of how many control and experimental untouched tibial plateaus of dogs were analysed. How less cartilage damage in the modified groove model was analyzed. Incorporate the elaborated results
- It is mentioned, line 263, “All three parameters are ….. histological findings”, here what are these 3 parameters? Mention it.
- Why results of force plate analysis, and cartilage integrity, chondrocyte activity, and synovial inflammation were evaluated on the surgically untouched tibial plateaus. Sufficient relevant information must be provided
- Explain why no notable changes were observed despite the presence of histological damage.
Discussion:
1)Mechanism of cartilage damage introduced by diffuse scratching must be elaborated along with references. Why this model is less suitable. Sufficient information must be incorporated.
2) How the two model varies. What is the degree of variability.
3) What is focal defects and how standardization of focal defects attempted
4) Elaborate “large grooves were created parallel to directional movement of the femoral condyles”, why ?
5) The results obtained must be discussed sufficiently by mentioning relevant Figs and tables
6) Discuss potential hormonal effects on OA progression and any recovery differences between male and female dogs?
Conclusion:
- In this study how many cases were considered, must be mentioned. No significant difference was observed in Force plate analysis. However, macroscopic analysis revealed damage in both original groove model and less damage in the modified groove model. Histologic assessments, PG analysis also revealed insignificant results between both the models
- This study is mainly performed by clinician. Many terms are needed to be clarified. At this stage the manuscript is very much unclear and needs to be clarified by providing details of methodology, appropriate labelling in the dog image in detail.
- Author in this study tried to proof that modified groove model is better than original groove model. However, the modified model needed to be verified with more experimental proof using more number of cases and control followed by systematic clear strategy, disadvantage/advantages of the generated model, potential recovery/ survivability of animal after surgery.
- Also it is very much unclear regarding the earlier data utilization of Original groove model
- Limitation of study must have been incorporated
Figs/Tables
- Label full-thickness cartilage grooves on the femoral condyles ,surgically untouched tibial plateaus of dog image.
- Patella, weight bearing region, cartilage grooves made in weight bearing region, femoral condyles, the menisci and tibial plateaus etc must be labelled in relevant Figs
- Abbreviation used in Figs/tables must be spelled as a footnote below each Fig/table legend
Other
- Abbreviation used (NSAIDs, DMOADs, PG, MRI,GW, HE, etc) must be spelled when appearing 1st time in the manuscript.
Comments on the Quality of English Language
moderate english improvement needed
Author Response
We would like to thank the reviewer for the time and effort to review our manuscript and for providing feedback to improve the paper. We have addressed items raised point by point. In these comments we refer to the adjusted lines, the numbers correspond to that of the redlined version of the manuscript (we have submitted this as a PDF file).
Reviewer 1:
Abstract:
Comment 1: “Information about earlier published article is not necessary in the abstract. Name of the methods used to evaluate needful parameters in the study must be mentioned.”
Response 1: Thank you for this comment. As we are presenting a modified version of the original groove model, we feel we need to provide some context on the model used. We have added the details on the methods used to evaluate the joint state, lines 26-27.
Comment 2: “Results must be crisp in the abstract section. Detail results must be provided in the result section of main manuscript and not in abstract”
Response 2: Thank you for the comment, we have adapted the results section and made it more focused as requested, hopefully to the satisfaction of the reviewer. lines 27-32.
Introduction:
Comment 3: “Prevalence of disease must be mentioned”
Response 3: We have added the most recent data on the prevalence of the disease including the reference; lines 46-48.
Comment 4: “Side effects of NSAIDs, corticosteroids, and DMOADs must be mentioned along with references”
Response 4: This line is more a reference that these animal models are often used to test OA related treatments including these types of drugs, there was no intention to discuss this in detail, especially as we described a new model and not discussing the effectiveness of these treatment options. As such we feel it is beyond the scope to include the details asked. We hope the reviewer agrees, otherwise we can add such details.
Comment 5: “Fig of Canine groove model showing surgically applied chondral damage to the weight-bearing cartilage of the femoral condyles may be generated and must be labelled appropriately for information and clarity in understanding.”
Response 5: When applying the grooves with the original groove model, the femoral condyles are not fully visible. As such we have not been able to generate photographs of the direct applied surgical damage of the original groove model.
Comment 6: “Mention anatomical differences between canine and human knees.”
Response 6: We have provided additional differences between canine and human knees and have expanded this paragraph of the introcuction in lines 55-64.
Comment 7: “How this model will be utilized for testing regenerative therapies. Please explain sufficiently”
Response 7: We have added more detail on potential utilization pf this model to the introduction, lines 104-105, as well in the discussion lines 384-386.
Comment 8: “What do the matrix turnover means that resembles human clinical OA. Incorporate information along with reference”
Response 8: In terms of pain and functional impairment (clinical OA), it is demonstrated that the dogs exhibit lameness and reduced joint function following the induction of OA using the groove model. We have added this information showing that these structural changes also result in clinical OA related changes reinforcing the validity of this model towards human OA. Lines 81-85
Comment 9: “Why progression to fully developed OA, in the ACLT model at 54 months post-surgery, takes more time? How much time ACLT model takes and how much time groove model Sufficient information must be provided along with references”
Response 9:
The ACLT model is the only canine OA model with a documented follow-up period as long as 54 months. Both the ACLT and Groove models are slow-progressing, resembling the human OA condition more closely than rapid-onset models in smaller animals (e.g., rodents) or chemically induced models. Therefore, we expect disease progression in the Groove model to be similar. However, this has not been demonstrated, primarily due to ethical and financial constraints, as allowing OA to progress to complete joint degeneration was unnecessary for the studies conducted. We have clarified this point in the introduction (lines 85–89).
Comment 10: “Further, it is mentioned (line 63) that groove model exhibits progressive cartilage degeneration alongside a gradual reduction in synovial inflammation. Its highly confusion. How reduction in synovial inflammation happens when there is progressive cartilage degeneration? Clarification needed.”
Response 10: We agree with the reviewer that this is confusing; we have adapted this line. Due to the initial operation, there is a clear increase in inflammation which lowers over time but there is still an increased, albeit low inflammation, of the synovial tissue while the cartilage damage progresses. Line 79
Comment 11: “What is diffuse chondral damage. Incorporate information along with references”
Response 11: Diffuse refers to widely spread or scattered over a large area, lacking a well-defined boundary or focal point. We have added the relevant reference in line 100.
Comment 12: “Line 49, References for the line “The dog is one of the best … knee OA” must be incorporated.”
Response 12: thank you, we have relocated the relevant reference that was cited at the end of the line 67.
Comment 13: “Line 56, References for the line “This ACLT model, …….joint instability” must be incorporated.”
Response 13: thank you, we have relocated the relevant literature that was cited at the end of the line 73.
Comment 14: “Reference for line 82 “Currently, there is a lack of suitable models for adequately evaluating cartilage repair strategies” must be mentioned.”
Response 14: Thank you, we have added the relevant literature, line 105.
Methods:
Comment 15: “Which method was used for animal sampling and what was the duration of experiment using the said models. Mention it.”
Response 15: We are unsure what the reviewer means by this question. Assuming it refers to the timing of the analysis or sample collection, we have added further details in the Methods section. The total duration of the experiment for both groups was 45 weeks, we have further clarified this in our response to comment 23.
Comment 16: “It is mentioned line 91 that the data of Original groove model (ref 10) was used as retrospective comparison. What were these data that were used to compare the current results. This data got published in the year 2015. Please clarify.”
Response 16: As these data were already gathered and to limited animal use and costs involved it was decided in consultation with our ethical committee to make a retrospective comparison. The data of the group used had an exact similar experimental set-up, including housing location, surgery team supplier of animals, as well the team who processed the samples obtained, compared to the modified model presented. As such this made the most fitting retrospective comparison.
Comment 17: ”It is mentioned, line 115 that the resulting cartilage fragments were left in the knee joint.Further, in modified groove model, the obtained cartilage pieces were removed from the joint. Why so? Is there any specific reason. Please clarify.”
Response 17: Thank you for this comment, we have added our rational on the original groove model that resulting cartilage fragments were small (<2mm) and sometimes still connected to the surrounding cartilage and therefore left in the knee joint in lines 141-142 and the rational on the modified groove model that obtained cartilage pieces were large(up to 2cm) and not connected to the cartilage, therefore due to foreseen interference we removed the fragments from the joint in lines 153-154.
Comment 18: ”How Maximum stance force (Fz) and braking force (Fymax) values were measured. Name the instrument, company name etc.”
Response 18: We have added the details on the instrument and company in the adjusted methods section of the manuscript; values were measured using a floor-mounted force plate, 0.6 × 0.9 m, set into a path 13 m long. A type Z4852C force plate, type 1681 cables, 5007 charge amplifiers and a type 5217 summation amplifier (Kistler) were used. The sampling rate was 100 Hz. This can be found in lines 195-198.
Comment 19: “What is scaffold implantation? Incorporate sufficient information along with references”
Response 19: Thank you for this comment, we have added an example to line 104-105.
Comment 20: “Briefly mention the Safranin- O/Fast green staining procedure, Alcian blue staining procedure, HE-stained, sections quantification of glycosaminoglycan (GAG) content, measurement of cartilage tissue proteoglycan levels etc along with references.”
Response 20: The techniques used are common histological and biochemical techniques to evaluate cartilage changes. Together with the restricted word count, we therefore provided limited details on the exact protocols. We added additional references to refer to more detailed protocols and originals resources for those interested. Lines 223 & 229.
Comment 21: “Mention company and country name for purchasing chemicals (such as papain, Alcian Blue, chondroitin sulfate etc), instruments (such as spectrophotometer, microscope etc)”
Response 21: We have added the relevant details in the methods section where applicable. Papain, P‐3125; Sigma-Aldrich; Saint Louis, MO; 25 mg/ml in 50 mm phosphate buffer, pH 6.5, containing 2 mmN‐acetyl cysteine and 2 mm Na2‐EDTA. Alcian Blue 8GX, A‐5268; Sigma-Aldrich; Saint Louis, MO; saturated in 0.1 m sodium acetate buffer, pH 6.2, containing 0.3 m MgCl2; 30 min, 37°C. Chondroitin sulfate: C4383; Sigma-Aldrich; Saint Louis, MO. Na235SO4, 14.8 kBq/200 μl, carrier‐free; NEX‐041‐H; DuPont; Wilmington, DE. These can be found in lines 229-235 and 239-240.
Comment 22: “References for dogs anesthesia must be incorporated.”
Response 22: Protocols are dog specific and in consultation with the veterinary team. We have added this to the manuscript, and we have added a reference where similar anesthesia was used for similar experiments, line 172-173.
Comment 23: “Mention why groups were monitored up to 45 weeks post-surgery. Incorporate information along with references”
Response 23: This follow-up period was selected to allow sufficient time for OA progression and to enable comparison with the retrospective group. It also aligns with the typical timeframe for applying regenerative treatments, for which this model was developed. We have added this rational to the method section, line 189-192.
Comment 24: “Mention why the results were analysed 35 and 45 days after post surgery”
Response 24: OA induction surgery introduces both operation-related pain and OA-related pain. Based on previous studies, surgical pain typically subsides within 3 to 5 weeks. To ensure objective measurements, we analyzed only the last 10 weeks of data, consistent with the retrospective group.
Comment 25: “What is nonparametric comparisons”
Response 25: The term nonparametric refers to statistical methods that do not assume a specific distribution for the underlying data. Unlike parametric methods, which require assumptions about parameters such as mean and variance (e.g., normal distribution), nonparametric techniques are more flexible and suitable for analyzing data that do not meet these assumptions, as typical found in explorative data. The statistical tests used are specific for this nonparametric data; e.g. Mann-Whitney U test and Wilcoxon's signed rank test
Comment 26: “How biochemically proteoglycan content was analysed. Briefly describe”
Response 26: As stated in line 227-229 cartilage tissue proteoglycan content from eight predefined samples across the tibial plateaus was determined. Glycosaminoglycans (GAGs), representative for proteoglycans, were extracted using papain, precipitated, and stained with Alcian Blue, and measured spectrophotometrically at 620 nm absorbance, with chondroitin sulfate as a reference (Sigma catalog no. C4383). The results were normalized to the wet weight of the cartilage explants and expressed in mg/g. We have added additional references, as mentioned under comment 20 to refer to more detailed protocols.
Comment 27: “Mention the importance of Mann-Whitney U test and Wilcoxon's signed rank test”
Response 27: The Mann-Whitney U test and Wilcoxon signed-rank test are two fundamental and commonly used nonparametric tests used in statistical analysis used in biomedical research. They are essential for analyzing data that do not meet the assumptions of normality required by parametric tests like the t-test. Their importance lies in their flexibility, robustness, and ability to handle skewed, ordinal, or small sample size data, as presented in the manuscript.
Outcome measures:
Comment 28: “Briefly mention how and why Force plate analysis were performed along with references”
Response 28: We provided more detail on why and how force plate analysis was performed, see methods section (line 195-198 and 201-205).
As a surrogate measure of pain/functional ability, limb loading during gait was evaluated by force plate analysis, values were measured using a floor-mounted force plate, 0.6 × 0.9 m, set into a path 13 m long. A type Z4852C force plate, type 1681 cables, 5007 charge amplifiers and a type 5217 summation amplifier (Kistler) were used. The sampling rate was 100 Hz. The dogs are guided to walk in a constant walking speed, both the front and hind leg of the same laterality must be placed subsequently on the force plate mounted flush to the surface. The force plate measures peak ground reaction forces in the 3 dimensions (x, y, and z). A single handler guided the dogs by leash over the force plate, at a constant walking speed of ~1+/0.2 meters/second (mean +/- SD). A successful run consisted of sequential, distinct paw strikes of the left and right hind limb. On average, 10 valid runs of each side were collected and ground reaction forces were averaged for each limb separately.
Comment 29: “Mention the criteria used macroscopic evaluation, the scores for cartilage and synovial tissue”
Response 29: The score is assigned based on the following criteria and is determined by the most severe pathology noted: 0 smooth surface, 1 roughened, 2 slightly fibrillated, 3 fibrillated, 4 damaged. Scores of the two observers were averaged (maximum of 4). This score was used as the representative score for each photograph and was used for statistical analysis. We have added this in the method section lines 214-217 & 254-258 which we have adapted including the reference for original detailed information.
Comment 30: “Name the biochemical assays performed using eight cartilage samples per tibial plateau and how proteoglycan data of each joint was obtained.”
Response 30: We refer kindly to the adapted method lines 227-236.
Comment 31: “What is contralateral joint and why it was used as internal control”
Response 31: The contralateral knee is the knee on the other side. As surgery is performed in the right knee, the contralateral knee refers to the left knee, which is surgically untouched, and serves as an internal non-OA control. Previous studies confirm its suitability, showing no significant differences from healthy control joints. Using an internal control reduces inter-animal variation, minimizing the number of animals required per experiment. We have clarified the latter in the method section, lines 177-182.
Results:
Comment 32: “What is stance and braking force and why these parameters were measured.Importance of these parameter in this study must be mentioned.”
Response 32: In force plate analysis, stance force refers to the ground reaction force measured during the stance phase of gait—when the limb is in contact with the ground. It reflects how much weight the animal places on that limb. Brake force (also called braking force) is the horizontal force acting opposite to the direction of movement during the early stance phase, when the limb decelerates the body. Together, these forces help assess limb function, load bearing, and gait abnormalities. Based on literature it is know that these forces, specifically the stance force are best related to the clinical features of OA in dogs as such there was a specific focus on these forces. We have added this rational with relevant literature in the method section, lines 207-286.
Comment 33: “In Cartilage integrity assay, mention surgery of how many control and experimental untouched tibial plateaus of dogs were analysed. How less cartilage damage in the modified groove model was analyzed. Incorporate the elaborated results”
Response 33: Cartilage samples for histological and biochemical analyses were obtained from predetermined locations on the weightbearing areas of the tibial plateau of both the operated and control joints. This way of tissue sampling is also extensively described in previous publications of the original Groove model. A such, this was also applied in the current study. We have adapted this in the methods section line 218-220.
Comment 34: “It is mentioned, line 263, “All three parameters are ….. histological findings”, here what are these 3 parameters? Mention it.”
Response 34: All three refers to the proteoglycan content, total PG release and release of newly formed PGs. We have adapted the line into “the proteoglycan content, total PG release and release of newly formed PGs” to clarify the three parameters, see line 326.
Comment 35: “Why results of force plate analysis, and cartilage integrity, chondrocyte activity, and synovial inflammation were evaluated on the surgically untouched tibial plateaus. Sufficient relevant information must be provided”
Response 35: The force plate analysis and synovial inflammation are different analysis which do not relate to the tibial plateaus. The cartilage integrity and chondrocyte activity were evaluated on the surgically untouched tibial plateaus to minimize influence from the direct applied surgical damage. We have provided this rational to the manuscript, lines 187-189.
Comment 36: “Explain why no notable changes were observed despite the presence of histological damage.”
Response 36: We have addressed this question specifically in different parts of the discussion. In short:
The modified groove model exhibits less degeneration compared to the original groove model for several reasons:
Removal of Loose Cartilage Bodies: In the modified model, the cartilage obtained from the grooves was completely removed, unlike in the original model where residual cartilage remained. Since loose cartilage bodies are known to cause pain and irritation, their removal likely reduced ongoing joint degeneration.
Reduction in Continued Degenerative Triggers: The absence of residual cartilage in the modified model eliminated an additional contributing factor to joint degeneration, allowing for a more stable evaluation of osteoarthritis (OA) progression and treatment options.
Less Articular Surface Incongruity: The modified groove model resulted in less disruption of the tibial plateau due to smoother, more parallel grooves along the natural movement direction of the femoral condyles. In contrast, the original model created rougher, disorganized grooves, leading to greater macroscopic cartilage damage.
Lower Synovial Inflammation: Although both models showed mild synovial inflammation, the original groove model exhibited significantly more macroscopic synovial tissue inflammation. This is likely due to the presence of loose cartilage pieces in the joint, which can trigger inflammatory responses.
Comparable but Less Severe OA Development: While both models displayed significant cartilage deterioration and similar osteoarthritis progression (as indicated by proteoglycan loss and retention), the modified model appears less sensitive and less prone to additional joint irritation.
Overall, the modified groove model reduces excessive degeneration by minimizing secondary damage triggers, leading to a more controlled assessment of osteoarthritis pathology and potential treatments.
Discussion:
Comment 37: ”1)Mechanism of cartilage damage introduced by diffuse scratching must be elaborated along with references. Why this model is less suitable. Sufficient information must be incorporated.”
Response 37: As described in the original publications, the pattern that is applied results in diffuse mild degeneration the cartilage on both the condyles as tibial plateau, especially the weight-bearing areas. This is a combination of initial cartilage damage applied by the grooving in combination with the usage and loading of the joint. These are general well-accepted principles in the OA field. These publications have been included in the manuscript, line 100.
Comment 38: “2) How the two model varies. What is the degree of variability.”
Response 38: Both models show comparable amount of variability, although this differs per parameter. Based on the previous studies in which the Groove model was used similar variability and effects (OA severity) were observed, which indicates that it is a relative robust model
Comment 39: “3) What is focal defects and how standardization of focal defects attempted”
Response 39: A focal defect in cartilage refers to a localized area of damage or loss of cartilage within a joint, typically affecting a specific, well-defined region rather than the entire cartilage surface. The grooves created in the modified Groove model resembles more a focal defect, as can be seen in figure 1d.
Comment 40: “4) Elaborate “large grooves were created parallel to directional movement of the femoral condyles”, why ?”
Response 40: The direction chosen was to ensure that the grooves could be made upon the weight-bearing area of the femoral condyles, to induce the OA process. If this is done in a non-weight bearing area there will be no development of OA due to lack of loading. While we discuss that this direction in hindsight might be less beneficial it was not feasible with the existing tools to do it otherwise. We have added this to the method section lines 151-153 and elaborate on the importance in the discussion line 415-417.
Comment 41: “5) The results obtained must be discussed sufficiently by mentioning relevant Figs and tables”
Response 41: We appreciate the reviewer’s comment. Upon review, we believe the main findings have been adequately discussed with reference to the relevant figures and tables. However, we have revisited the discussion section to ensure all key results are clearly linked to the corresponding data presentations and have made minor adjustments where needed for clarity.
Comment 42: “6) Discuss potential hormonal effects on OA progression and any recovery differences between male and female dogs?”
Response 42: In the modified Groove model group, only one of the five dogs was male, limiting our ability to assess sex-based differences in OA progression. Based on our expertise and previously published data using the Groove model, we do not expect significant differences in OA progression related to sex.
Conclusion:
Comment 43: “In this study how many cases were considered, must be mentioned. No significant difference was observed in Force plate analysis. However, macroscopic analysis revealed damage in both original groove model and less damage in the modified groove model. Histologic assessments, PG analysis also revealed insignificant results between both the models”
Response 43: The number of animals included in each group (n=9 for the original Groove model and n=5 for the modified Groove model) is specified, and appropriate statistical tests were applied to each parameter, as described in the Methods section. While some outcomes did not reach statistical significance—particularly in the force plate and histologic assessments—we believe the combined trends across macroscopic and histological findings still offer meaningful insight into differences between the models. We have reviewed the text to ensure clarity around sample sizes and the interpretation of non-significant results.
Comment 44: “This study is mainly performed by clinician. Many terms are needed to be clarified. At this stage the manuscript is very much unclear and needs to be clarified by providing details of methodology, appropriate labelling in the dog image in detail.”
Response 44: We thank the reviewer for this observation. In response, we have revised the manuscript to improve overall clarity, particularly by refining the methodology section and enhancing image labelling. While we recognize that reviewer opinions may differ, we have aimed to balance detailed explanations with readability, ensuring the content is accessible to both clinical and scientific audiences.
Comment 45: “Author in this study tried to proof that modified groove model is better than original groove model. However, the modified model needed to be verified with more experimental proof using more number of cases and control followed by systematic clear strategy, disadvantage/advantages of the generated model, potential recovery/ survivability of animal after surgery.”
Response 45: We appreciate the reviewer’s comment and would like to clarify that our aim was not to prove the modified Groove model as superior, but rather to assess its suitability for evaluating repair strategies targeting focal cartilage defects. While the modified model may offer advantages for specific applications, we agree that further validation including larger sample sizes, appropriate controls, and a more systematic evaluation of outcomes is necessary to fully establish its utility. We have clarified this intent in the revised manuscript.
Comment 46: “Also it is very much unclear regarding the earlier data utilization of Original groove model”
Response 46: As discussed in item number 16, we have tried to clarify this.
Comment 47: “Limitation of study must have been incorporated”
Response 47: We have added an additional paragraph to the discussion section, discussing additional limitations. Lines 432-452
Figs/Tables
Comment 48: “Label full-thickness cartilage grooves on the femoral condyles ,surgically untouched tibial plateaus of dog image.”
Response 48: We have added labeling of the femur and tibia to the image to provide more clarity.
Comment 49: “Patella, weight bearing region, cartilage grooves made in weight bearing region, femoral condyles, the menisci and tibial plateaus etc must be labelled in relevant Figs”
Response 49: We thank the reviewer for this helpful suggestion. In response, we have added labels for the femur, indicated by a * in the image and the tibia, indicated by a + to improve anatomical clarity. Unfortunately, the patella and menisci are not visible in the current view, and accurately indicating the weight-bearing region would require a sagittal plane image, which is not available in this figure.
Comment 50: “Abbreviation used in Figs/tables must be spelled as a footnote below each Fig/table legend”
Response 50: Where relevant we have made adaptations.
Other
Comment 51: “Abbreviation used (NSAIDs, DMOADs, PG, MRI,GW, HE, etc) must be spelled when appearing 1st time in the manuscript.”
Response 51: We have added these abbreviations to the manuscript. GW, AB and SCM are the initials of the observers.
Reviewer 2 Report
Comments and Suggestions for Authors
The manuscript presents a study comparing a modified canine groove model of osteoarthritis (OA) with the original groove model. The modified model involves full-thickness cartilage grooves using a 3-mm biopsy punch, aiming to assess its suitability for evaluating cartilage repair strategies. The study examines force plate analysis, cartilage integrity, chondrocyte activity, and synovial inflammation in five mongrel dogs, comparing the results with a previously studied cohort using the original model. The findings suggest that both models induce OA features, with the modified model demonstrating less severe cartilage damage and synovial inflammation. The authors propose that the modified model is useful for studying cartilage repair interventions.
Abstract: clearly defines the study objectives, methodology, and key findings. The comparison between models is well-articulated.
Introduction: provides a comprehensive background on OA, its complexity, and the need for animal models. Limitations of the ACLT model are highlighted and the need for the groove model is explained comprehensevily. However, the distinction between the original and modified models is not emphasized early enough.
Materials and Methods: provides detailed procedural descriptions, including surgical techniques, perioperative management, and statistical analyses. Ethical approval and adherence to guidelines are well-documented.
The small sample size (n=5 for the modified model) raises concerns about statistical power, please point that out in the manuscript.
Lack of randomization or blinding procedures in animal allocation may introduce bias. How was bias minimized.
Please add more details on force plate analysis standardization (e.g., how inter-trial variability was handled) would be beneficial.
Results: data is presented in a structured and clear manner.
Discussion: effectively places the findings in the context of existing OA models. Differences between the two groove models and their implications for cartilage repair studies are acknowledged.
Force plate analysis results lack discussion on variability and trends over time.
The potential influence of external factors (e.g., individual animal variability, environmental factors) is not considered.
Conclusion: key findings are summarized concisely.
Figures and Tables: well organized and clear. Labels and legends are appropriate. Some tables and figures (e.g., force plate results) lack error bars or confidence intervals.
General Statement: minor revision
This manuscript presents an interesting and potentially valuable study, but several key issues need to be addressed before publication. By addressing the above mentioned points, the manuscript will be significantly improved and better positioned for publication in your journal.
sincerely
Author Response
We would like to thank the reviewer for the time and effort to review our manuscript and for providing feedback to improve the paper. We thank the reviewer for the positive feedback on the writing and presenting the data of this manuscript (comments 1, 2, 4,8, 9, 12). We have addressed the remaining concerns point by point. In these comments we refer to the adjusted lines, the numbers correspond to that of the redlined version of the manuscript (we have added this as a pdf file to that attachments).
Comment 1: “The manuscript presents a study comparing a modified canine groove model of osteoarthritis (OA) with the original groove model. The modified model involves full-thickness cartilage grooves using a 3-mm biopsy punch, aiming to assess its suitability for evaluating cartilage repair strategies. The study examines force plate analysis, cartilage integrity, chondrocyte activity, and synovial inflammation in five mongrel dogs, comparing the results with a previously studied cohort using the original model. The findings suggest that both models induce OA features, with the modified model demonstrating less severe cartilage damage and synovial inflammation. The authors propose that the modified model is useful for studying cartilage repair interventions.”
Comment 2: “Abstract: clearly defines the study objectives, methodology, and key findings. The comparison between models is well-articulated.”
Comment 3: “Introduction: provides a comprehensive background on OA, its complexity, and the need for animal models. Limitations of the ACLT model are highlighted and the need for the groove model is explained comprehensevily. However, the distinction between the original and modified models is not emphasized early enough.”
Response 3: We have adapted the introduction accordingly to make the difference between the two model more clearly, lines 107-109.
Comment 4: “Materials and Methods: provides detailed procedural descriptions, including surgical techniques, perioperative management, and statistical analyses. Ethical approval and adherence to guidelines are well-documented.”
Comment 5: “The small sample size (n=5 for the modified model) raises concerns about statistical power, please point that out in the manuscript.”
Response 5: We agree with this comment and have added this this to the limitations as discussed in lines 445-447.
Comment 6: “Lack of randomization or blinding procedures in animal allocation may introduce bias. How was bias minimized.”
Response 6: Unfortunately, due to a retrospective comparison, randomization was not possible. By maintaining all other factors identical and assessors blinded we tried to minimize this bias.
Comment 7: “Please add more details on force plate analysis standardization (e.g., how inter-trial variability was handled) would be beneficial.”
Response 7: We have adjusted the description of the analysis to clarify this measurement, see method section lines 201-205.
Comment 8: “Results: data is presented in a structured and clear manner.”
Comment 9: “Discussion: effectively places the findings in the context of existing OA models. Differences between the two groove models and their implications for cartilage repair studies are acknowledged.”
Comment 10: “Force plate analysis results lack discussion on variability and trends over time.”
Response 10: As results show very limited effect on force plate analysis, we have kept this limited. We have added the challenges around this type of measurement to the limitations of the study. Lines 433-438
Comment 11: “The potential influence of external factors (e.g., individual animal variability, environmental factors) is not considered.”
Response 11: thank you, we will add this to additional limitations section.
Comment 12: “Conclusion: key findings are summarized concisely.”
Comment 13: “Figures and Tables: well organized and clear. Labels and legends are appropriate. Some tables and figures (e.g., force plate results) lack error bars or confidence intervals.”
Response 13: As we want to show the individual values as well the mean in those figures we chose not to show error bars in the figure to keep them readable. The mean values including the variation are also given in the table.
Comment 14: ”General Statement: minor revision. This manuscript presents an interesting and potentially valuable study, but several key issues need to be addressed before publication. By addressing the above mentioned points, the manuscript will be significantly improved and better positioned for publication in your journal.”
Round 2
Reviewer 1 Report
Comments and Suggestions for Authors
Title: The modified canine groove model of osteoarthritis.
Revised comments: Author had satisfactorily incorporated most of the comments response in the revised manuscript. However, few minor comments as mentioned below is needed to be incorporated
- In response to comment 4, it is necessary to add side effect of currently available drugs for OA disease, in order to justify the needfulness of the said study with new model
- Please clarify response to comment 10. Line 79 “Due to the initial operation, there is a clear increase in inflammation which lowers over time but there is still an increased, albeit low inflammation, of the synovial tissue while the cartilage damage progresses” is not mentioned in the revised manuscript.
Comments on the Quality of English Language
Moderate improvement on English language is needed
Author Response
Comment 1: Revised comments: Author had satisfactorily incorporated most of the comments response in the revised manuscript. However, few minor comments as mentioned below is needed to be incorporated.
Response 1: We thank the reviewer for this positive feedback and are happy to comply with the additional minor comments.
Comment 2: In response to comment 4, it is necessary to add side effect of currently available drugs for OA disease, in order to justify the needfulness of the said study with new model.
Response 2: We have added side effects to NSAIDs, corticosteroids and DMOADs with relevant literature in lines 64-70 of the adjusted (redlined) manuscript.
Comment 3: Please clarify response to comment 10. Line 79 “Due to the initial operation, there is a clear increase in inflammation which lowers over time but there is still an increased, albeit low inflammation, of the synovial tissue while the cartilage damage progresses” is not mentioned in the revised manuscript.
Response 3: In the original manuscript we mentioned a gradual reduction in synovial inflammation, this reduction is when compared to post-operatively. However there is still low grade synovial inflammation, which is why we reformulated it this way. We have now added this explanation to the manuscript in lines Lines 78-80 of the adjusted (redlined) manuscript.